# Blood-Flow Restriction Resistance Exercise for Older Adults with Knee Osteoarthritis: A Pilot Randomized Clinical Trial

**DOI:** 10.3390/jcm8020265

**Published:** 2019-02-21

**Authors:** Sara A. Harper, Lisa M. Roberts, Andrew S. Layne, Byron C. Jaeger, Anna K. Gardner, Kimberly T. Sibille, Samuel S. Wu, Kevin R. Vincent, Roger B. Fillingim, Todd M. Manini, Thomas W. Buford

**Affiliations:** 1Department of Medicine, University of Alabama at Birmingham, Birmingham, AL 35205, USA; saharper@uabmc.edu (S.A.H.); lmroberts@uabmc.edu (L.M.R.); 2Center for Exercise Medicine, University of Alabama at Birmingham, Birmingham, AL 35205, USA; 3Department of Aging and Geriatric Research, University of Florida, Gainesville, FL 32610, USA; andrew.laynephd@gmail.com (A.S.L.); akgardner@ufl.edu (A.K.G.); ksibille@ufl.edu (K.T.S.); tmanini@ufl.edu (T.M.M.); 4Department of Biostatistics, University of Alabama at Birmingham, Birmingham, AL 35205, USA; bcjaeger@uab.edu; 5Department of Biostatistics, University of Florida, Gainesville, FL 32610, USA; samwu@biostat.ufl.edu; 6Department of Orthopaedics and Rehabilitation, University of Florida, Gainesville, FL 32610, USA; vincekr@ortho.ufl.edu; 7Department of Community Dentistry and Behavioral Science, University of Florida, Gainesville, FL 32610, USA; rfillingim@dental.ufl.edu

**Keywords:** aging, osteoarthritis, pain, function, blood-flow restriction

## Abstract

In a pilot randomized clinical trial, participants aged ≥60 years (*n* = 35) with physical limitations and symptomatic knee osteoarthritis (OA) were randomized to 12 weeks of lower-body low-load resistance training with blood-flow restriction (BFR) or moderate-intensity resistance training (MIRT) to evaluate changes in muscle strength, pain, and physical function. Four exercises were performed three times per week to volitional fatigue using 20% and 60% of one repetition maximum (1RM). Study outcomes included knee extensor strength, gait speed, Short Physical Performance Battery (SPPB) performance, and pain via the Western Ontario and McMaster Universities OA Index (WOMAC). Per established guidance for pilot studies, primary analyses for the trial focused on safety, feasibility, and effect sizes/95% confidence intervals of dependent outcomes to inform a fully-powered trial. Across three speeds of movement, the pre- to post-training change in maximal isokinetic peak torque was 9.96 (5.76, 14.16) Nm while the mean difference between groups (BFR relative to MIRT) was −1.87 (−10.96, 7.23) Nm. Most other directionally favored MIRT, though more spontaneous reports of knee pain were observed (*n* = 14) compared to BFR (*n* = 3). BFR may have lower efficacy than MIRT in this context—though a fully-powered trial is needed to definitively address this hypothesis.

## 1. Introduction

Osteoarthritis (OA) is the most common cause of disability among persons over 60 years of age, affecting approximately 30–50% of this population [1,2]. Given the increasing population of older adults worldwide, the prevalence of OA is expected to increase dramatically in coming years [3,4]. Among the important clinical implications of OA in older adults is the contribution of OA to declining physical function [5,6,7]. Specifically, OA in the weight-bearing joints contributes to the majority of functional decline, with knee OA being the most common and most limiting [8,9,10]. Additionally, older adults with knee OA typically have greater difficulty performing tasks important for independence compared to their non-affected peers [11,12]. Older adults with risk factors for knee OA have benefited from BFR resistance training [13,14,15]. Thus, interventions may address factors which contribute to decline in function among older adults with knee OA. 

Skeletal muscle weakness, particularly in the quadriceps muscles, contributes to knee OA-related functional decline [7,8]. Traditional interventions aimed at increasing skeletal muscle strength commonly involve high-load or high-intensity resistance training, and may also result in joint pain for persons with OA due to high-compressive forces [16,17]. Subsequently, joint pain may present a barrier to high-load resistance training among some older persons with knee OA [18]. To limit joint pain, current recommendations for adults with OA include low- to moderate-intensity resistance training. However, this approach may limit skeletal muscle strength gains [18,19]. Currently, there is limited evidence regarding alternative intervention strategies for older adults with knee OA [20,21]. However, novel strategies may reduce pain and improve physical function for older adults with knee OA.

As suggested previously [18], low-load resistance training performed in concert with blood flow restriction (BFR), is a potential intervention for lowering training related joint pain. BFR training involves applying external compression to the exercising limb and mildly restricts blood flow to the activated skeletal muscle. BFR resistance training has been independently validated as a viable option for increasing skeletal muscle strength and mass in some older adults, including those with risk factors for knee OA [13,14,15]. The objectives of this pilot randomized clinical trial (RCT) were (1) to assess the safety and efficacy of BFR training in older adults in knee OA, and (2) to assess the feasibility for a fully-powered RCT comparing BFR training to a clinically-recommended moderate-intensity resistance training (MIRT) intervention among older adults with knee OA. 

## 2. Materials and Methods

### 2.1. Study Overview

The study design and objectives were described in detail previously [18]. Briefly, this study is a two-arm randomized, single-masked pilot trial to evaluate the safety and efficacy of BFR training for improving muscle strength and OA-related symptoms among older adults with knee OA. Participant safety was overseen by a comprehensive study team—including the principal investigator, study physician, study staff, and an appointed Data and Safety Monitoring Board. Prior to enrollment, all participants provided written informed consent approved by a university Institutional Review Board. Additionally, the study was registered at www.clinicaltrials.gov (NCT02132715). 

The single-masked design is an accepted procedure for exercise-based studies as indicated by the Consolidated Standards of Reporting Trials (CONSORT) Group [22,23] where study staff conducting the assessments were masked to the intervention assignment. Study intervention and study assessments also occurred in separate physical locations and intervention groups were conducted at different times to prevent contamination bias between groups. Statistical analyses were performed by a biostatistician who remained masked to intervention assignments throughout. 

### 2.2. Participants

Men and women from the Gainesville, Florida area with knee OA were recruited and then randomized into either MIRT or BFR exercise training interventions. Recruitment involved a targeted approach of direct mailings, newspaper advertisements, and other community approaches. Eligibility criteria included (1) ≥60 years of age, (2) objective functional limitations, (3) no participation in regular resistance training, and (4) symptomatic knee OA. The presence of knee OA was defined by (1) radiographic evidence of osteophytes, (2) pain classification > grade 0 on Graded Chronic Pain Scale, [24], and (3) bilateral standing anterior–posterior radiograph demonstrating Kellgren and Lawrence grade ≥ 2 of the target knee [25]. X-rays were read by a board-certified radiologist or physiatrist. Exclusionary criteria included (1) contraindications to tourniquet use, including those with peripheral vascular disease, (2) resting systolic blood pressure (SBP) >160 or <100 mm Hg, diastolic blood pressure (DBP) >100 mm Hg; absolute contraindications to exercise training [25,26] or any other medical conditions that preclude safe participation.

### 2.3. Assessments

Unilateral isokinetic strength of knee extensors on the limb with knee OA was assessed via a dynamometer (Biodex Medical Systems, New York, NY, USA) as previously published [27,28]. If knee OA was present in both limbs, the limb with higher self-reported pain was used. Walking speed was assessed by asking participants to walk at their usual pace over 10 laps of a 40 m course as described previously [29,30]. Immediately following the 400 m walk, current pain was assessed with a visual analog scale (VAS). Lower-extremity function was assessed via the Short Physical Performance Battery (SPPB) as described previously [31,32].

The Late Life Function and Disability Instrument (LLFDI) was used to report self-assessed physical function [33,34]. The instrument includes 16 tasks covering a broad range of disability indicators assessing frequency of task performance and perceived limitation. Scores are tallied on a scale of 0 to 100 where higher scores indicate higher levels of function. The Western Ontario and McMaster Universities Osteoarthritis Index (WOMAC) was used to assess knee-related pain. The WOMAC is a multidimensional, self-administered functional-health status instrument for patients with lower-limb OA with shown validity and sensitivity to treatment effects in patients with knee pain [35]. Knee pain questions were self-assessed on a 0–4 scale where participants assessed activity difficulty (0 = none; 1 = mild, 2 = moderate, 3 = severe, 4 = extreme).

Commercially available enzyme-linked immunosorbent assays (ELISA) kits were used to determine serum concentrations of N-terminal peptide of procollagen type III (P3NP, MyBioSource, San Diego, CA, USA), tumor necrosis-like weak inducer of apoptosis (TWEAK, R&D Systems, Minneapolis, MN, USA), and insulin-like growth factor (IGF-1, R&D Systems, Minneapolis, MN, USA). Concentrations of target proteins were identified using the colorimetric method at an optical density of 450 nm with a microplate reader (Biotek, Winooski, VT, USA). Intra-assay coefficients of variation for each assay were determined for each duplicate for all participants and resulted in a mean coefficient of variation of 3.4%.

### 2.4. Interventions

Both study groups participated in center-based, supervised resistance exercise training three times a week. Participants began with a brief warm-up followed by lower-body strength training and ending with flexibility exercises and balance training. All resistance exercises were performed using standard isotonic resistance training equipment (Life Fitness, Schiller Park, IL, USA). 

Participants were introduced to exercises with a familiarization session to provide participants time to acclimate to the exercise machines and learn proper technique. Participants’ initial one repetition maximum (1 RM) for four standard lower limb exercises (leg, press, leg extension, calf flexion, leg curl) was used to determine starting weights. Both groups retested 1RM assessments at the 3 week (9th session), 6 week (18th session), 9 week (27th session), and 12 week (36th session) assessment visit. After each 1RM testing, exercise weights were recalculated for all participants to reflect their new 1RM.

Participants in the MIRT group performed four lower-extremity exercises (leg press, leg extension, leg curl, and calf flexion) at 60% of 1RM, following exercise guidelines for seniors with OA [36,37,38]. The low-load BFR resistance training group performed the same four lower-extremity exercises at 20% of 1RM with the addition of external compression applied to the proximal thigh of both legs. Compression was applied according to published tourniquet guidelines [34,39] and sustained throughout the duration of each exercise, including in between sets, by pneumatic cuffs (TD 312 calculating cuff inflator, Hokanson, Bellevue, WA, USA) [39,40]. Following each lower-extremity exercise set, cuffs were deflated for the rest periods between the different exercises. Cuff pressure was individualized to each participant according to the equation [pressure mm Hg = 0.5 (SBP) + 2(thigh circumference) + 5] adopted from prior BFR and tourniquet research literature [41,42,43]. To ensure equal metabolic work between groups, exercises were performed to volitional fatigue. 

### 2.5. Safety

Safety of participants was our highest priority through the pilot RCT. Potential adverse events were explained in detail to participants during the consent process, and participants were instructed to notify study staff immediately if an event occurred. Events were documented and subsequently monitored per IRB and DSMB guidelines. In addition, clinical blood tests were performed at baseline, week 6, and week 12 during the intervention to monitor potential adverse biochemical responses to the interventions. 

### 2.6. Statistical Analysis

Linear mixed models were applied to estimate change over time, overall and by study group, for each continuous outcome [44,45]. A group-by-time interaction was used to estimate intervention-based changes from baseline, and additional adjustment was included for baseline pain score, exercise group, and study visit (baseline, week 6, and week 12). As this study was not powered to detect statistical differences in outcomes (e.g., *p*-value < 0.05), estimated mean differences are presented with 95% confidence intervals (CIs) according to published guidelines for reporting findings from pilot studies [46,47]. A secondary efficacy analysis was conducted among participants with ≥ 80% exercise attendance and attended baseline and week 12 study visits (see Appendix A). 

## 3. Results

### 3.1. Participants

Participants (*n* = 35) were randomized to study interventions (BFR: *n* = 16, MIRT: *n* = 19) (Figure 1). At the baseline assessment, characteristics of the two study groups were similar (Table 1). However, there was a substantial difference in self-reported knee pain between study groups via a visual analog scale indicating a potentially greater baseline knee pain among the MIRT group. 

### 3.2. Retention, Adherence, and Safety 

A total of *n* = 33 (94.2%) participants were retained and completed the study. One participant from each study group withdrew their consent. Three participants in each group discontinued participation in the exercise intervention at some point during the intervention period but remained in the trial. Still, adherence to the exercise interventions was ≥80% in each study arm (BFR: 81.4%; MIRT: 83.0%). Regarding safety, a total of 34 post-randomization adverse events or spontaneous adverse reports of knee pain were observed (13 BFR; 21 MIRT). Of these, 21 were deemed related or possibly related to the study (6 BFR; 15 MIRT). The majority of these events were related to knee pain (*n* = 14), and the BFR group had less of these reports (*n* = 3) than the MIRT group (*n* = 11). A total of five serious adverse events were observed (2 BFR; 3 MIRT), with only one (BFR group) deemed related or possibly related to the study. Within group changes were similar among clinical chemistry and hematology panels, except for white blood cell count (BFR relative to MIRT: −0.79 (−1.44, −0.13) cells/L) and sodium 1.93 (0.10, 3.75) mmol/L (Appendix A).

### 3.3. Exercise Training Volume and 1RM

Aggregate training volume (repetitions x pounds used) for each exercise are shown in Table 2. Rating of perceived exertion (RPE) per exercise session measured via the Borg scale [48] was 7.3 ± 0.5 for BFR compared to 8.1 ± 0.5 for MIRT. Overall, pre- to post-training changes in 1RM for the four training exercises were as follows (mean and 95% CI): leg press 72.29 (40.47, 105.11) lbs, leg extension 41.34 (27.50, 55.19) lbs, calf flexion 75.16 (45.64, 104.68) lbs, and leg curl 17.67 (7.61, 27.72) lbs. Differences in post-training changes in 1RM between groups were as follows (BFR relative to MIRT): leg press −50.81 (−117.22, 15.60) lbs, leg extension −26.60 (−54.94, 1.74) lbs, calf flexion −30.66 (−91.05, 29.73) lbs, and leg curl −16.46 (−36.05, 3.13) lbs. 

### 3.4. Primary and Secondary Outcomes

Data from the intent-to-treat analysis are reported below. Across both groups, the pre- to post- training change in the mean composite knee extensor peak torque (Figure 2A) was 9.96 (5.76, 14.16) Nm. The mean change between groups for mean composite knee extensor peak torque at week 12 was (BFR relative to MIRT) −1.87 (−10.96, 7.23) Nm. The intent-to-treat analyses for knee extensor strength had positive changes at week 12 for both groups across three speeds of movement 60°/s, 90°/s, and 120°/s (Appendix A). The mean change across 12 weeks in the 400 m walk gait speed was −0.03 (−0.08, 0.01) m/s. The difference in post-training changes in 400 m walk gait speed between groups was −0.01 (−0.11, 0.09) m/s (Figure 2B). Furthermore, across both groups, the pre- to post-training change in SPPB score was 0.47 (−0.03, 0.97) points. Between groups at post-training, the mean difference in SPPB was −0.66 (−1.74, 0.42) points (Figure 2C). Efficacy analyses for mean composite knee extensor strength, 400 m walk, and SPPB displayed similar results (Appendix A). 

The training change in LLFDI Frequency Total for both groups (pre- to post-training) was −0.14 (−2.23, 1.94) points. Difference between groups at post-training was −0.79 (−6.76, 5.17) points (Figure 3A). Pre- to post-training change in the LLFDI Limitation Total was 4.36 (0.06, 8.72) points and change at post-training between groups was as −6.60 (−18.99, 5.79) points (Figure 3B). The WOMAC pain subscale difference across both groups was −0.81 (−2.04, 0.42) points while between group difference was 0.24 (−2.51, 2.98) points (Figure 3C). In general changes observed in efficacy analyses were similar, though group changes were more similar for the WOMAC pain sub-scale (Appendix A). 

Across both groups, the mean change in total lean mass from pre- to post-training was 0.40 (−0.61, 1.40) kg. Between groups (BFR relative to MIRT), the mean change was −1.10 (−3.44, 1.24) kg. Total body fat percentage across both groups changed −1.02 (−0.13, −1.91) % and the between groups difference at post-training for total body fat percentage was 1.12 (−0.90, 3.14) %. Across both groups, the post-training change in lower body lean mass was 0.71 (1.07, 0.36) kg, with a between groups difference of −0.44 (−1.26, 0.39) kg (Figure 4). 

### 3.5. Biomarkers

The mean change in serum P3NP across groups was −0.63 (−1.28, 0.02) mg/dL and the difference in serum P3NP post-training between groups was −0.98 (−2.39, 0.44) mg/dL (Figure 5A). Serum TWEAK changed 21.70 (−90.16, 133.56) mg/dL from pre- to post-training. Post-training serum TWEAK difference between groups was −92.70 (−306.14, 120.74) mg/dL (Figure 5B). Across both groups, pre- to post-training change in serum IGF-1 was −0.06 (−0.13, 0.02) mg/dL. Difference in post-training IGF-1 between groups was −0.04 (−0.19, 0.11) mg/dL (Figure 5C). Biomarker changes were similar when evaluated using efficacy analyses (Appendix A).

## 4. Discussion

This pilot RCT investigated the safety and feasibility of resistance exercise with blood flow restriction (BFR) compared to moderate-intensity resistance training (MIRT) for older adults with knee osteoarthritis (OA). As outlined previously [18] and in line with best practices for pilot studies [49], the purpose was to refine and finalize elements critical to conducting a future, fully-powered randomized, controlled trial to test the extent to which BFR training in older adults with knee OA modulates skeletal muscle strength, physical function, and pain compared to MIRT. 

Among the most important aspects of the trial was to determine the safety and feasibility of implementing the protocol in the target population. In total, only seven adverse events (excluding expected reports of knee pain) related or possibly related to the study were reported – with only three of these coming from the BFR group. Though one serious adverse event was reported in the BFR group, in aggregate the intervention appeared to be safe and possibly result in fewer reports of knee pain than traditional MIRT. Notably, the MIRT group had a total of 11 reports of knee pain compared to the three reported incidents of knee pain in the BFR group—an important observation given that one of the primary goals of this pilot RCT was to improve strength while minimizing knee pain in this population. Still, serious adverse events—including rhabdomyolysis [50]—have been reported in the literature, thus tremendous care should still be taken to minimizing risks and ensuring safety among older persons engaging in BFR exercise.

Regarding feasibility, an important component of our pilot RCT among older adults with knee OA was reviewing participant acceptability of the BFR intervention [51]. As described previously, both treatment groups had similar, high adherence percentages (above 80%) for both the assessment visits and exercise sessions, implying that the participants in the BFR group tolerated the BFR intervention. Moreover, technical aspects of feasibility including randomization masking procedures were successfully implemented and the protocol appears to be implementable on the larger scale.

Among the challenges experienced in the trial was participant recruitment, commonly among the most challenging aspects of study execution [52]. Common barriers to recruitment in RCTs include both the recruitment of participants to screening visits as well as overly-restrictive screening criteria which may overly exclude participants from participating. Regarding the former, the geographic location of the study may have limited overall recruitment to the trial. The location was in a relatively small metropolitan area—possibly indicating the need for a larger area or possibly the need for more sites. Moreover, the availability of numerous other age-related studies available may have contributed to a large number of individuals withdrawing prior to randomization given the necessary time requirements for the trial. Regarding screening criteria, over one-third of participants who were screened for the study were not randomized with the primary exclusion criterion being that they did not have objectively-measured physical limitations. Given the objective was to utilize the interventions to ultimately improve function, it was important to keep this criterion to prevent a ceiling effect among enrolled participants who would not benefit. However, one potential option for consideration may be utilizing a recruitment criterion specific to skeletal muscle strength given the focus on strength in the trial. 

Importantly, the serum biomarkers evaluated hold potential for explaining, in a fully-powered trial, any potential differences between BFR and MIRT groups. For example, increases in circulating P3NP is associated with skeletal muscle growth as well as with muscle repair and fibrosis [53]. Meanwhile, IGF-1 is well-known to be associated with increased protein synthesis and lean mass [54]. Additionally, lower TWEAK concentrations are associated with skeletal muscle hypertrophy and greater force production [55]. Though the data should not be over-interpreted in this pilot, the data for serum P3NP appear to be in line with clinical data as, in absolute terms (i.e., not statistically significant), concentrations were higher among the MIRT group at week 12 which would be expected to be associated with improvements in muscle mass/function. Serum TWEAK and IGF-1 appear to be more equivocal. These outcomes may be considered for a larger trial, but a larger panel of potential outcomes may also provide more mechanistic insight. 

Other important considerations for future studies include significant staff demand for the BFR intervention as well as the comparison of exercise volume between groups. Because of potential safety concerns if BFR is performed inappropriately, participants in the BFR group required an individual dedicated staff interventionist to administer inflation and deflation of the tourniquet as well as, monitoring the participant for safety during the exercise protocol. Thus, staff requirements were higher for this group compared to MIRT. Regarding exercise volume, BFR studies are inherently difficult to match both contractile and metabolic work given the differing loads performed. We chose to have participants go to volitional failure to achieve similar metabolic work. Despite this objective and re-testing 1RM throughout the study, the aggregate training volume for leg press was greater for the MIRT group. However, metabolic work is inherently different for matching workload (sets x reps) when factoring in the cuff restriction. Therefore, standardizing to volitional fatigue may be a viable option to minimize the workload differences. 

## 5. Conclusions

In summary, this pilot RCT indicated that BFR was a safe and feasible alternative for older adults with knee OA and these results provide necessary data to plan a full-scale RCT comparing MIRT and BFR. These data add to prior literature suggesting BFR as a potential therapeutic strategy for older adults and/or those with osteoarthritis [13,15]. As a pilot, the findings should not be over-interpreted but do provide supportive data indicating the potential for BFR as an alternative training regimen to improve pain and function among older adults with knee OA. Although, BFR may potentially have lower efficacy than MIRT, a fully-powered RCT is needed to formally address this hypothesis. 

## Figures and Tables

**Figure 1 jcm-08-00265-f001:**
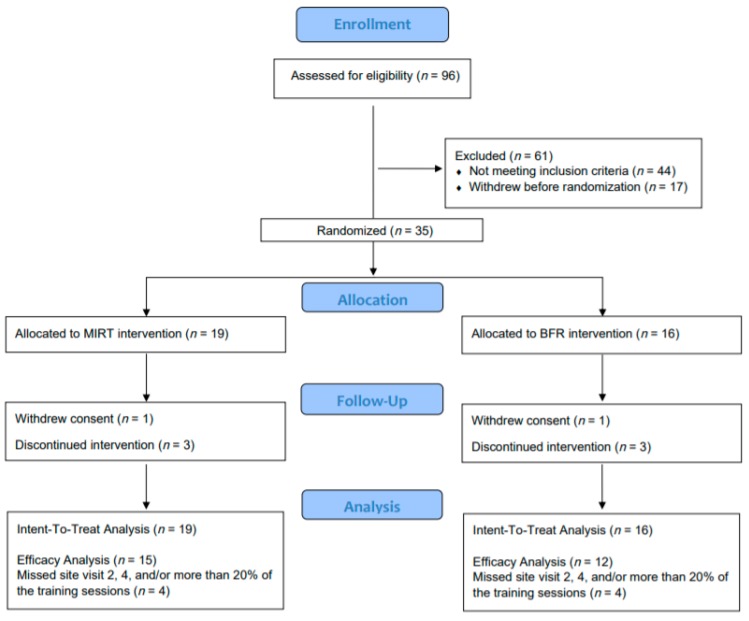
Flow diagram of study progress in the Consolidated Standards of Reporting Trials (CONSORT) Group.

**Figure 2 jcm-08-00265-f002:**
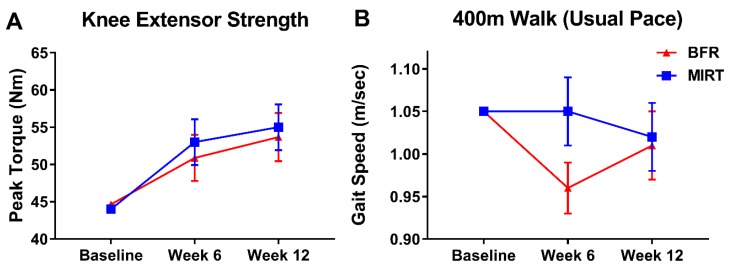
(**A**–**C**) Objective measures of physical function were evaluated from baseline to week 12 by mean composite unilateral knee extensor peak torque (Nm) across three speeds of movement (60, 90, 120°/s) (**A**), 400 m walk usual pace gait speed (**B**), and Short Physical Performance Battery (**C**). Values indicate estimated marginal mean ± SEM. Abbreviations: BFR—blood-flow restriction, MIRT—moderate-intensity resistance training.

**Figure 3 jcm-08-00265-f003:**
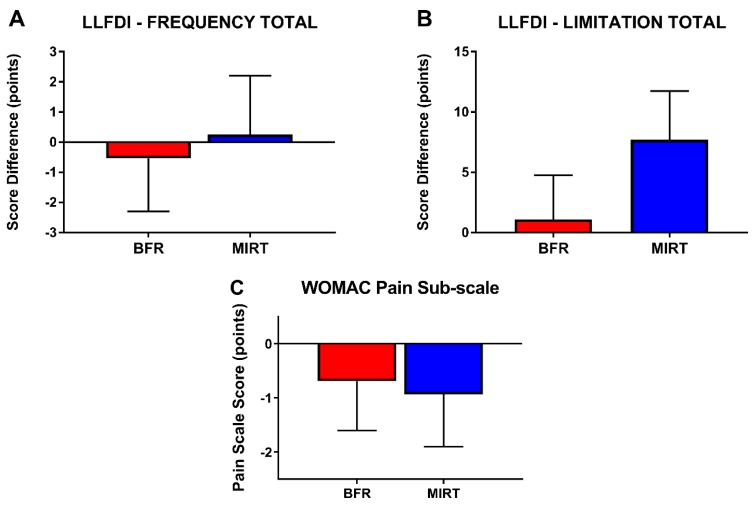
(**A–C**) Subjective measures of pain and function were evaluated from baseline to week 12 via reported the group mean difference for Late Life Functional and Disability Instrument (LLFDI) Total Disability Frequency (**A**), Total Disability Limitation (**B**) where a higher reported score represents better performance, and WOMAC pain sub-scale (**C**). Values indicate estimated marginal mean ± SEM. Abbreviations: BFR—blood-flow restriction, MIRT—moderate-intensity resistance training, WOMAC—Western Ontario and McMaster Universities Osteoarthritis Index.

**Figure 4 jcm-08-00265-f004:**
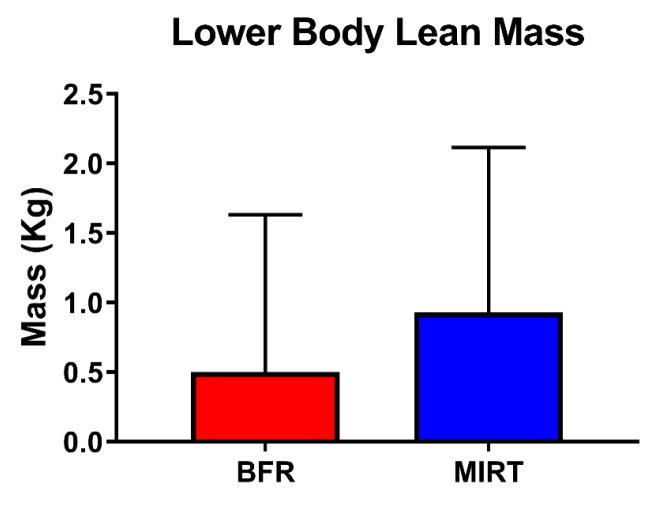
Total lower body lean mass was evaluated from baseline to week 12 via reported the group mean difference. Values indicate estimated marginal mean ± SEM. Abbreviations: BFR—blood-flow restriction; MIRT—moderate-intensity resistance training.

**Figure 5 jcm-08-00265-f005:**
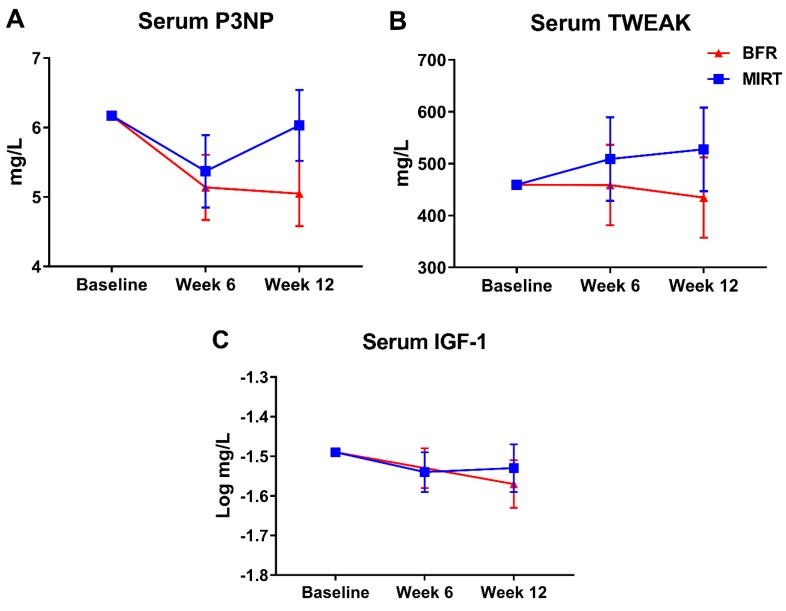
(**A**–**C**) Serum biomarkers of myogenic activity and collagen turnover including N-terminal peptide of procollagen type III (P3NP) (**A**), tumor necrosis-like weak inducer of apoptosis (TWEAK) (**B**), and insulin-like growth factor (IGF-1) (**C**). Values indicate estimated marginal mean ± SEM. Abbreviations: BFR—blood-flow restriction, MIRT—moderate-intensity resistance training.

**Table 1 jcm-08-00265-t001:** Participant demographic baseline characteristics

	MIRT (*n* = 19)	BFR (*n* = 16)
Age, years	69.1 ± 7.1	67.2 ± 5.2
Sex, Female	15 (78.9%)	10 (62.5%)
Race, White	73.7%	81.2%
Ethnicity, Hispanic	6.2%	6.2%
Body Mass Index, kg/m^2^	29.8 ± 5.3	31.7 ± 5.9
Systolic blood pressure, mm Hg	132 ± 19	126 ± 15
Diastolic blood pressure, mm Hg	80 ± 10	73 ± 6
Kellgren and Lawrence score, grade	2.9 ± 0.8	2.8 ± 0.8
Visual analog pain scale, mm	28.1 ± 19.9	11.1 ± 11.1
400 m walk gait speed, m/s	1.01 ±0.11	1.04 ± 0.12
WOMAC pain subscale, points	7.23 ± 4.87	6.19 ± 3.04
60 deg/s peak torque extension, Nm	45.3 ± 19.3	52.3 ± 12.1
90 deg/s peak torque extension, Nm	41.5 ± 17.1	49.7 ± 12.4
120 deg/s peak torque extension, Nm	35.1 ± 14.7	46.2 ± 13.3
Total SPPB, points	10.2 ± 1.9	10.4 ± 1.9
Leg Press 1RM, lbs	130.1 ± 63.4	139.7 ± 38.0
Leg Extension 1RM, lbs	90.6 ± 41.6	92.5 ± 20.3
Leg Curl 1RM, lbs	94.8 ± 23.1	101.3 ± 21.1
Calf Flexion 1RM, lbs	147.1 ± 56.3	160.3 ± 68.4

Values are mean ± SD, or *n*, percentage. SPPB = short physical performance battery, 1RM = one repetition maximum.

**Table 2 jcm-08-00265-t002:** Total volume, repetitions, and weight for each exercise by group.

	BFR	MIRT
Leg press total volume, lbs.	793 ± 495	1709 ± 908
Leg press repetitions	452.04 ± 369.82	376.94 ± 166.04
Leg press weight, lbs.	885.05 ± 666.26	2436.22 ± 1712.89
Leg extension total volume, lbs.	357 ± 165	639 ± 363
Leg extension repetitions	213.13 ± 163.63	180.75 ± 154.44
Leg extension weight, lbs.	511.33 ± 345.41	1051.31 ± 1099.43
Leg curl total volume, lbs.	900 ± 377	931 ±303
Leg curl repetitions	367.88 ± 283.26	310.67 ± 160.05
Leg curl weight, lbs.	1014.70 ± 616.26	1595.05 ± 955.81
Calf flexion total volume, lbs.	1314 ± 921	1779 ± 858
Calf flexion repetitions	570.07 ± 473.99	383.57 ± 224.55
Calf flexion weight, lbs.	1025.27 ± 886.93	2485.47 ± 1507.80

Lower body exercise total volume, repetitions and weight on average per training session attended. Values are reported as mean ± SD.

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
