# Peer review of "Blood-Flow Restriction Resistance Exercise for Older Adults with Knee Osteoarthritis: A Pilot Randomized Clinical Trial"

_jcm, 2019, doi:10.3390/jcm8020265_

Reviewer 1 Report

The authors revised the content of the article one by one according to the reviewer's comments. Some of the enhanced serum biomarkers including the N-terminal peptide of procollagen type III (P3NP), tumor necrosis-like weak induccer factor (TWEAK) and insulin-like growth factor (IGF-1) were discussed, and the clinical significance between MIRT and BFR, and citing relevant literature, has met the journal level.

Author Response

The attached document we present the specific details addressed by each reviewer along with lines line numbers where changes appear. 

Reviewer 2 Report

The manuscript by Harper et al. made a pilot randomized clinical trial on the aged patients with knee osteoarthritis (OA), which compared 12 weeks of lower-body low-load resistance training with blood-flow restriction (BFR) or moderate-intensity resistance training (MIRT).  To evaluate the patients changing in muscle strength, pain, and physical function. Some concerns were noted in the presentation of findings, and interpretation of the results. Please consider the following revisions to the manuscript.

First, the study does not report on the presence or absence of adverse events in the clinical trial. Although injury resulting from this type of training seems rare, which the risks of adverse events may be exacerbated in all clinical populations, the muscle damage is common in BFR exercise and is necessary for training effects/adaptations, the possible risks of rhabdomyolysis occurring during BFR exercise may be heightened in cases of muscular disuse atrophy.  

Second, the authors performed the serum biomarkers to explain the potential differences between BFR and MIRT groups. The authors should make more specific discussing closing to the results in the last part of the manuscript. And there doesn't describe any details about how did they develop the ELISA on the serum. Please add it in the MM.

In the discussion, the authors don’t make any discussion about  BFR safety outcomes, which is the main topic of this paper. 

Author Response

In the attachment we present specific details to address each comment along with line numbers where changes appear. 

This manuscript is a resubmission of an earlier submission. The following is a list of the peer review reports and author responses from that submission.

Round  1

Reviewer 1 Report

The authors randomly selected aged ≥60 years (n=35) with good physical condition and symptomatic knee osteoarthritis (OA) in a randomized clinical trial to be divided into 12-week hypotension and low-load resistance training, blood flow restriction. blood-flow restriction (BFR) or moderate-intensity resistance training (MIRT) to assess muscle strength, pain, and changes in body function. Comparing the necessary data for MIRT and BFR, BFR is considered to be a safe and viable alternative for knee OA elderly. The results of this study have clinical application value in the treatment of BFR in elderly and/or steoarthritis patients.  

Serum biomarkers for detecting myogenic activity and collagen turnover, including N-terminal peptide of procollagen type III (P3NP), tumor necrosis-like weak inducer of apoptosis (TWEAK) and insulin-like growth factor (IGF-1), the value resulting discussion should strengthen the clinical significance between the MIRT and BFR in the text. 

The data in the tables and figures that are statistically significant should be marked with p value.